# Autism Spectrum Disorders and the Gut Microbiota

**DOI:** 10.3390/nu11030521

**Published:** 2019-02-28

**Authors:** Antonella Fattorusso, Lorenza Di Genova, Giovanni Battista Dell’Isola, Elisabetta Mencaroni, Susanna Esposito

**Affiliations:** Pediatric Clinic, Department of Surgical and Biomedical Sciences, Università degli Studi di Perugia, 06132 Perugia, Italy; antonella.fattorusso@libero.it (A.F.); lory.digenova@gmail.com (L.D.G.); Giovanni.dellisola@gmail.com (G.B.D.); elisabetta.mencaroni@ospedale.perugia.it (E.M.)

**Keywords:** Autism spectrum disorder, brain, gut microbiota, microbiota transfer therapy, probiotic

## Abstract

In recent years, there has been an emerging interest in the possible role of the gut microbiota as a co-factor in the development of autism spectrum disorders (ASDs), as many studies have highlighted the bidirectional communication between the gut and brain (the so-called “gut-brain axis”). Accumulating evidence has shown a link between alterations in the composition of the gut microbiota and both gastrointestinal and neurobehavioural symptoms in children with ASD. The aim of this narrative review was to analyse the current knowledge about dysbiosis and gastrointestinal (GI) disorders in ASD and assess the current evidence for the role of probiotics and other non-pharmacological approaches in the treatment of children with ASD. Analysis of the literature showed that gut dysbiosis in ASD has been widely demonstrated; however, there is no single distinctive profile of the composition of the microbiota in people with ASD. Gut dysbiosis could contribute to the low-grade systemic inflammatory state reported in patients with GI comorbidities. The administration of probiotics (mostly a mixture of *Bifidobacteria*, *Streptococci* and *Lactobacilli*) is the most promising treatment for neurobehavioural symptoms and bowel dysfunction, but clinical trials are still limited and heterogeneous. Well-designed, randomized, placebo-controlled clinical trials are required to validate the effectiveness of probiotics in the treatment of ASD and to identify the appropriate strains, dose, and timing of treatment.

## 1. Introduction

Autism spectrum disorder (ASD) is a complex group of developmental disorders characterized by impaired social interactions and communication together with repetitive and restrictive behaviours. Among ASDs, autism represents the primary type. Recent epidemiological studies reported prevalence rates in the general population of 58–67/10,000 [1] and an incidence of 1 in 68 children in 2012 in the United States, suggesting that ASD affects many families and represents a serious public health problem. Various factors have been associated with the development of ASD, including both genetic and environmental factors such as nutritional deficiencies or overloads, exposure to viruses, errors during embryonic neural tube closure, dysfunctional immune systems and allergies [2,3]. The genetic background of ASD is complex and includes genes involved in the development of the central nervous system (CNS) [3]. More than 100 genes and genomic regions have been implicated in the aetiology of ASD, and a greater number of genes (350–400) may confer autism susceptibility [4,5]. Accumulating evidence has highlighted the role of environmental factors, which are now believed to play a much more important role in ASD than previously assumed [6,7].

Even if the exact etiopathogenesis of ASD is poorly understood, in recent decades, research has pointed to the interaction between the gut microbiota and the brain in patients with autism or other neuropsychiatric diseases. A considerable number of subjects with ASDs have significant gastrointestinal dysfunctions, particularly altered bowel habits and chronic abdominal pain, that accompany their neurological alterations [8]. The gastrointestinal (GI) symptoms of individuals with ASD seem to correlate strongly with the severity of their ASD [9].

The microbiota, a microbial community of trillions of microorganisms and at least 1000 different bacterial species, few eucaryotic fungi and viruses, that covers every surface of the human body, plays a contributory role in many infections, immune-mediated disorders, rheumatologic diseases and disorders of the nervous system [10,11,12]. The microbiome, instead, is the collection of the whole genome sequences of those microorganisms, consisting of more than 5,000,000 genes [13,14]. Gut microbiota is strictly linked to the chronological age of each individual and modulates host physiology and metabolism through different mechanisms. Microbiota role in health and disease is as crucial as is complex. Alterations in normal commensal gut microbiota result in an increase in pathogenic microbes that deranges both microbial and host homeostasis. This microbial imbalance is known as dysbiosis and it has been widely reported as a key contributor to the multiple system dysregulation observed in the pathogenesis of cardiovascular metabolic, neuroimmune and neurobehavioural conditions [15,16,17]. The alteration of the gut microbial community contributes to the pathophysiology of many gastrointestinal conditions, such as inflammatory bowel diseases, functional bowel disease, food allergies, obesity and liver diseases [18]. Furthermore, recent studies have suggested that alterations in the gut microbiota composition (i.e., dysbiosis) in children with ASD may contribute to both gastrointestinal and CNS symptoms [19,20]. Thus, research has focused on changes in the gut microbiota as a risk factor in individuals who are genetically predisposed to ASD; these changes in the gut microbiota are thought to influence the risk for ASD by influencing the immune system and metabolism [21,22].

In recent years, several studies have shown significant changes in the composition of the gut microbiota in children with ASD [23,24] and have suggested that GI symptoms in ASD may be a manifestation of the underlying inflammatory process [25]. In particular, dysbiosis is associated with a disruption of the mucosal barrier that leads to increased intestinal permeability of exogenous peptides of dietary origin or neurotoxic peptides of bacterial origin (such as lipopolysaccharide (LPS)) [22] and the production of inflammatory cytokines [25]. Indeed, the gut microbiota and the related metabolites play a crucial role in the so-called “gut-brain axis” [26,27,28,29], a physiological bidirectional complex network of communication between the brain and the gut [27,30,31]: The disruption of neural, endocrine and metabolic mechanisms that are involved in gut-CNS signalling seems to be involved in neuropsychiatric disorders, including autism and ASDs [22]. In this paper, we review evidence of dysbiosis in ASD, focusing on the possible link between gastrointestinal disorders, inflammation and neurobehavioural symptoms in autistic children. Finally, we discussed the current therapeutic approaches. The originality of our manuscript is the analysis of the necessity to consider or not the importance of gastrointestinal disorders in children with ASD. PubMed was used to search for all of the studies published over the last 15 years (from 1 January 2003 to 31 December 2018) using the key words “autism”, “gut” and “microbiota” or “microbiome”. Approximately 250 articles were found. Only those published in English and that investigated overall neurological disorders, specifically autism spectrum disorders, were included in the evaluation.

## 2. The Gut Microbiota

The human gut is inhabited by several trillion microorganisms that live in a symbiotic relationship with the host and account for approximately 1 kg of the weight of the gut [32]. The collection of all the microorganisms that live in the human gut, such as many different species of known bacteria, viruses, fungi, protozoa and archaea, is termed the microbiota [33]. The genome of the microbiota, including both genes and gene products, is defined as the microbiome [34]. The healthy adult gut microbiota is composed of 4 major phyla that together account for more than 90% of the total bacterial population [35]: *Bacteroidetes* (Gram negative such as the *Bacteroides* and *Prevotella* genera), *Firmicutes* (Gram positive aerobic and anaerobic bacteria such as *Lactobacillus*, *Clostridium* and *Ruminococcus*), *Proteobacteria* (e.g., *Enterobacter* species) and *Actinobacteria* (e.g., *Bifidobacterium*), followed by the minor phyla *Fusobacteria* and *Verrucomicrobia* [35].

The partnership between microorganisms and the host is essential for health and survival. First, the gut microbiota offers a barrier against the proliferation of pathogenic organisms and works synergistically to metabolize toxins, drugs and dietary compounds and to provide essential nutrients [36]. The functions of the gut microbiome include the production of metabolites such as butyrate or lactic acid that can have beneficial effects on the host due to their anti-inflammatory, anti-tumorigenic and antimicrobial properties [37]. Furthermore, recent evidence has shown that the microbiome is involved in the maturation and functionality of the host adaptive immune system [38]. The development and function of the CNS are also influenced by the production of microbial metabolites [35]. An experimental animal study showed that the absence of the microbiota during early life increases transcriptional pathways in the amygdala, a brain region involved in emotions, anxiety and social behaviours [39].

The composition of the gut microbiota varies widely from birth to adult age. Recent data suggest that the initial colonization of the foetus gut may occur before birth during intrauterine life via placental colonization [40]; then, further colonization occurs during delivery after the passage through the vagina [41] or, in the case of caesarean section, after contact with environmental microbes [35]. Other postnatal factors, such as antibiotic therapy, infections, breast-feeding vs. formula feeding, stress, diet and host genetics, determine the composition of the microbiome [42,43,44].

## 3. Dysbiosis in Autism Spectrum Disorder

A maternal high-fat diet during pregnancy alters the microbiota in neonates and might be associated with ASD in humans [45]. Breast-feeding is associated with a lower risk of ASD if continued for 6 months, while formula-fed infants show a greater representation of *Clostridium difficile* in the gut [46,47]. Antibiotic treatments, even if undertaken for a brief period, may induce long-lasting alterations in the gut microbiota, both in humans and in animal models [48,49]. Yassour et al. demonstrated that children treated with antibiotics during the first 3 years of life have different gut microbiome compositions [50], while Korpela and colleagues showed that a long-lasting alteration in the gut microbiota in children after a course of macrolide antibiotics may be associated with obesity and asthma [48]. Gut dysbiosis is reported in several conditions, such as immunological defects, Crohn’s disease, obesity, inflammatory bowel disease (IBD) and abnormal behaviours in children (including those with ASD) [27,38,51,52].

Alterations in the composition of the microbiota and its metabolites have been demonstrated both in animal models of ASD and in ASD children as well as in those with pervasive developmental disorder-not otherwise specified (PDD-NOS) [22,28,44,53]. In mice, the administration of the antiepileptic drug valproic acid (VPA) to the female during pregnancy induces autism-like social behaviours and differences within the *Firmicutes* and *Bacteroidetes* phyla in the offspring [54]. The enteric microbiome of children with autism is different from that of neurotypical siblings and healthy controls, and differences have also been noticed between individuals with autism disease (AD) and the other ASD that do not meet all the diagnostic criteria of AD (e.g., PDD-NOS) [22]. Increased microflora and reduced microbial diversity characterize the ASD gut microbiota; this combination of factors can lead to the overgrowth of harmful bacteria contributing to the severity of autistic symptoms [55,56].

By pyrosequencing the gut microbiome from faecal samples, lower levels of the phylum *Firmicutes* with a relatively higher abundance of *Bacteroidetes* have been reported [23]. *Bacteroidetes* are shorty-chain fatty acid (SFA)-producing bacteria, and their metabolites, especially propionic acid, may influence the CNS and autism behaviour by modulating the gut-brain axis [23]. De Angelis and colleagues demonstrated that the abundance of *Firmicutes* is lower in the faecal samples of autistic children than in faecal samples of healthy children, but no differences have been noted between the abundance in individuals with PDD-NOS and controls [22]. Decreased levels of the genus *Bifidobacterium*, which could have a protective role in autism through its anti-inflammatory properties, have also been reported, as well as reduced levels of *Prevotella, Coprococcus* and *Veillonellaceae*, which are responsible for carbohydrate digestion and fermentation [56,57]. *Lactobacillus, Clostridium, Desulfovibrio, Caloramator, Alistipes, Sarcina, Akkermansia, Sutterellaceae* and *Enterobacteriaceae* have been found to be more abundant in children with ASD and in the sibling controls than in healthy children [22,24,55,58]. *Desulfovibrio*, an anaerobic bacillus that is resistant to common antibiotics such as cephalosporins, is more common in individuals with ASD [23,58,59]. Autistic children had significantly more ear infections and were given greater amounts of antibiotics than control children [58], and this may lead to an overgrowth of *Desulfovibrio* species that produce important virulent factors (i.e., lipopolysaccharides) involved in the pathogenesis of autistic social behaviours [60]. Duodenal biopsies taken from ASD patients with GI problems revealed an increased level of the genus *Sutterella*, which is associated with mucosal metabolism, and these data were confirmed by another study of stools from ASD children [61,62]. However, the differences in gut microbiome composition found by Finegold in 2010 did not appear to be statistically significant between the sibling controls and the several autistic subjects. Since there is a great amount of evidence of multiple cases of ASD in families, it has been proposed that autistic children could transmit their faecal microbiota to their siblings or mates, which concurs with the development of autism in predisposed children [23]. Among the *Firmicutes* phylum, the overgrowth of the genus *Clostridium* in autistic children has been documented in several studies [23,57,61,63,64]. Parracho et al. reported a higher incidence of *Clostridium histolyticum* in autistic children than in the control group of healthy children, and the abundance of *Clostridium perfringens* in the stools of autistic children was confirmed by several other studies [24,57,63,64]. The possible role of *Clostridium* in the etiopathogenesis of autism was proposed for the first time in 1998 by Ellen Bolte, a mother of a child with regressive autism who observed the onset of neurobehavioural changes after repeated courses of antibiotics and consequent chronic diarrhoea [65]. Furthermore, the hypothesis of *Clostridium* as a potential risk factor for ASD was supported by a study in which children with regressive autism were treated with a 6-week oral course of vancomycin (an antibiotic used against *Clostridium*), which resulted in a significant improvement of both neurobehavioural and gastrointestinal symptoms [66]. However, relapsing gastrointestinal and neurobehavioural symptoms occurred gradually after treatment interruption, possibly because of the spores of *Clostridium* that were resistant to vancomycin and could turn into infective forms later [65,67]. *Clostridiaceae* are also able to produce some metabolites, such as phenols, p-cresol and indole derivates, that are potentially toxic to humans [68]; these metabolites were detected in faecal samples of children with PDD-NOS and in even greater amounts in samples from children with AD [24]. Interestingly, a recent review postulated the role of glyphosate (GLY), an environmental pesticide, in the pathogenesis of autism by increasing the growth of the toxin-producing Clostridia [69]. Indeed, *Clostridium perfringens* and *Clostridium botulinum* are known to be highly resistant to GLY, unlike other beneficial bacteria such as *Bifidobacterium* and *Lactobacillus*. Thus, environmental levels of GLY may deleteriously influence the gut-brain axis and contribute to the pathogenesis of autism through the alteration of the microbiome and the production of *Clostridium* toxins [69].

Because there is evidence that autistic individuals have maldigestion and malabsorption problems, a recent study analysed the microbiome of the duodenal mucosa of subjects with autism and found no differences in microbiome oral bacterial diversity compared to the healthy control group, as well as no difference in disaccharidase activity [70]. The oral microbiota has been recently investigated, but no differences have been found in the richness and diversity of the microbiota in salivary samples of individuals with AD and healthy children. However, a minor but statistically lower bacterial diversity was observed in toddlers with ASD compared to controls [71]. In particular, a low prevalence of *Prevotella*, a commensal microorganism involved in the metabolism of saccharides and in the biosynthesis of vitamins [72], was detected in salivary and dental samples of individuals with ASD [71]. As autistic patients are thought to have impaired carbohydrate digestion [61], restoring the relative deficiency of *Prevotella* may have a therapeutic potential for ASD symptoms [71].

Dysbiosis in ASD involves not only bacterial species but also yeasts, as reported in recent studies [24,73,74,75]. Gastrointestinal *Candida albicans* is two times more abundant in toddlers with ASD than in normal individuals and can release ammonia and other toxins, inducing autistic behaviours [74,75]. However, the role of fungi in ASD has not been clearly elucidated, and more studies with large samples are needed.

In summary, although many studies have been published on dysbiosis in the gut microbiota of individuals with ASD, there is still little consensus on the exact composition of the gut microbiome that is specific to individuals with ASDs, and some studies have described opposing results. The possible reason for these opposing results may be the lack of homogeneity in terms of age, diet, pharmacological treatment, geographic area, comorbidities, and the severity of neurobehavioural and gastrointestinal symptoms among the patients enrolled. For example, the faecal microbiota of younger people contains proportionally fewer *Bacteroidetes* compared to that of older people [76], and the microbiome profile can vary depending on the section of the GI tract from which the sample was taken [59]. Therefore, to correctly compare and evaluate findings on the gut microbiota in ASD, studies on homogeneous patient groups are needed.

## 4. Gastrointestinal Abnormalities in Autism Spectrum Disorder and the “Inflammation Hypothesis”

The hypothesis of a connection between gut microbiota disruption and ASD originated from the evidence that children with ASD may have GI disturbances. A wide range of GI abnormalities have been reported, such as diarrhoea, constipation, vomiting, feeding problems, reflux and abdominal pain [25]. According to two recent studies, approximately 40% of ASD patients complain of GI symptoms [77,78], and according to a clinical meta-analysis, functional GI disorders occur more often in ASD children than in healthy children [79]. It has been hypothesized that the presence of an altered microbiota associated with GI problems in a child with a genetic predisposition for ASD may facilitate the expression of an autistic phenotype or increase the severity of neurobehavioural symptoms [80]. Indeed, children with both ASD and GI disturbances may display more severe anxiety, irritability and social withdrawal compared with those without GI disturbances [81]. The severity of neurobehavioural symptoms seems to increase the risk of having GI problems [77] and vice versa [9]. In the CHARGE (CHildhood Autism Risks from Genetics and Environment) study, the incidence of GI abnormalities was associated with more consistent autism behaviours, as measured by the Aberrant Behaviour Checklist. On the other hand, autism severity measured by the Autism Treatment Evaluation Checklist (ATEC) was correlated with a higher GI severity index [79]. There is increasing evidence that GI disturbances in autistic children may be related to gut dysbiosis due to the inflammatory state [82,83,84,85,86], and the abnormal faecal microbiota in both patients with inflammatory bowel diseases and in ASD children with GI disturbances suggests this correlation.

Although there is still little consensus, some studies have demonstrated different expressions of inflammation markers in children with ASD [87,88,89,90], and many recent reviews are pointing to the correlation between inflammation and immune dysfunction in ASD children with GI comorbidities [25,81,85]. By comparing the transcriptome profiles of gut biopsies from children with ASD, ulcerative colitis (UC) and Crohn’s disease (CD), a group of researchers showed that the transcriptional profile of ileal and colonic tissues in the children with ASD was similar to that of individuals with inflammatory bowel disease [82]. However, other studies have not found an increase in faecal calprotectin, lactoferrin, secretory IgA or elastase in ASD patients compared to typical children [44,53,83]. Nevertheless, infiltration of lymphocytes, monocytes, natural killer cells and eosinophils has been observed in tissues from gut biopsies of children with autism as well as in individuals with other diseases such as immunodeficiency and food allergies [25]. An excess accumulation of advanced glycation end products (AGEs) has been reported in autism brains. Through their interaction with their putative receptor RAGE, AGEs can promote neuroinflammation, oxidative stress and neuronal degeneration. However, Boso and colleagues have suggested systemic inflammation as a result of the disfunction of the AGE-RAGE axis in autistic children, with higher levels of RAGE (i.e., receptors for advanced glycation end products) and their proinflammatory ligand S100A9 in the blood of those patients [84]. A recent study has shown that astrocytes derived from ASD children have higher levels of proinflammatory cytokines compared to control-derived neurons [85], and this could contribute to the altered development of neurons and synapses in children with ASD [7].

Behind this “inflammation hypothesis”, there is the idea that a disruption of the gut epithelial barrier, which is physiologically involved in controlling the transit of molecules from the GI tract by tight junctions, can lead to altered gut permeability [86]. The microbiota and its metabolites are crucial in maintaining epithelial barrier integrity; therefore, dysbiosis in ASD patients may alter gut permeability [91]. This condition, called “leaky gut” [21], may allow the passage of bacteria, toxins such as LPS and metabolites that activate the immune response and induce an inflammatory state into the bloodstream [86]. The activated immune system releases inflammatory cytokines and chemokines, which can modulate the CNS and contribute to the pathogenesis of autism by influencing the early stages of brain development [5]. Increased plasma levels of proinflammatory cytokines such as interleukin (IL)-1B, IL-6, IL-8, IL-12p40, tumour necrosis factor (TNF)-alpha, and transforming growth factor (TGF)-beta as well as hyperactive cellular immune responses in children with ASD have been reported, and these immune/inflammatory profiles may be related to the severity of the ASD-related neurobehavioural symptoms [92,93]. Hsiao and colleagues found a significant deficit in intestinal barrier integrity in mouse models of ASD, as reflected by the increased translocation of FITC-dextran across the intestinal epithelium and into circulation [91]. In addition, higher intestinal permeability in autistic children has been demonstrated by the mean lactulose/mannitol test in patients and in their first-degree relatives [53]. ASD children also have higher plasma levels of the gut permeability-modulating protein zonulin than typical toddlers, and the increase in zonulin levels seems related to the severity of autistic symptoms [85]. In contrast, other reports have not found differences in intestinal permeability in autistic toddlers [44,94,95,96], suggesting that gut barrier disruption may not be a marker of autism in all cases but only in children with ASD and GI problems. An interesting recent study by Rose and colleagues included four groups of children and distinguished between ASD children with irregular bowel habits (ASDGI), ASD children without GI symptoms (ASDNoGI), typically developing children with GI symptoms (TDGI) and typically developing children without current or previous GI symptoms (TDNoGI) [81]. After TLR-4 stimulation, compared to the ASDNoGI group, the ASDGI group showed increased plasma levels of IL-5, IL-15, and IL-17 and a lower production of the regulatory cytokine TGFb1 [81], which is associated with worse behaviour scores [97]. Moreover, the ASDGI and TDGI groups had different microbiome compositions, suggesting that dysbiosis in children with ASD is independent of the presence of GI symptoms [81], confirming the idea that the gut and systemic inflammation state may contribute to pathogenesis, especially in ASD children with GI comorbidities.

Moreover, gut dysbiosis may influence the CNS and modulate emotions, anxiety and behaviour. The bidirectional communication between the gut and brain through neuroendocrine and neuroimmune pathways, autonomic nervous systems and microbial toxin production represents the gut-brain axis [90].

First, as discussed above, the gut microbiota contributes to the control of gut epithelial permeability [7,91,93].

Secondly, the intestinal microbiome plays an important role in the maturation of the host immune system by modulating the innate and adaptive immune system, especially regulatory T-cells, which prevent inflammation [36]. Dysbiosis in ASD children leads to an activation of the immune system and the release of inflammatory cytokines that regulate the CNS through the vagal system [7,98,99,100].

Third, the gut microbiota can send signals to the CNS via the ENS or via the afferent fibres of the vagus nerve, both directly and through the release of neurotransmitters in the bloodstream [4].

The administration of *Lactobacillus rhamnosus* in mice stimulated the transcription of γ-aminobutyric acid (GABA) receptors through the vagal nerve, inducing behavioural and psychological responses, but this effect was reversed after vagotomy [101]. Several bacterial species can send signals to the CNS via the vagus nerve [4,7]. In top-down signalling, the autonomic nervous system and hypothalamus-pituitary-adrenal axis influence the gut microbiota as a neuroendocrine pathway [36].

Finally, the gut microbiota produces metabolites such as short-chain fatty acids (SCFAs), phenol compounds, and free amino acids (FAA), which are thought to play a critical role in ASD [7].

## 5. The Emerging Role of the Serotonin Pathway in the Autism Spectrum Disorder Gut-Brain Axis

In animal models, germ-free (GF) mice show higher levels of monoamines (noradrenaline, dopamine, serotonin) and lower levels of BDNF and anxiety-like behaviours than conventional mice [19,102,103]. An interesting recent study points to the role of the neurotransmitter serotonin (5-hydroxytryptamine, 5-HT) as a link for the gut-brain axis in ASD [104]. In addition, 5-HT is thought to be involved in the development of both the CNS and the ENS [105].

Hyperserotonemia in children with ASD has been demonstrated since the 1970s [106,107], and a correlation with gastrointestinal symptoms have recently been found by Marler and colleagues [108]. Because 90% of whole-blood 5-HT is synthesized by intestinal enterochromaffin (EC) cells, it has been postulated that higher levels of serotonin in ASD children may be caused by a gastrointestinal 5-HT hypersecretion [104]. Israelyan and Margolis studied serotonin transporter (SERT) variants and their correlations with hyperserotonemia in ASD children and incorporated the most common variant, the SERT Ala56 mutation, in a mouse model [104]. However, further investigations are needed to better understand whether 5-HT agonism may rescue behavioural abnormalities, and human studies are required as well.

The causes of elevated levels of serotonin in ASD patients are not only genetic. Infections, gastrointestinal disorders and immune system impairment are also thought to be involved [109,110]. Interestingly, recent animal studies have found altered microbiome compositions in ASD mouse models that were associated with GI disorders and increased intestinal production of 5-HT [111,112], supporting a connection between enteric serotonin production and dysbiosis. A clinical study performed on children with ASD and GI disorders revealed that, compared to children with ASD and no GI symptoms, the children with ASD and GI disorders exhibited a higher prevalence of *Clostridiales* species on their gut mucosa, which was associated with elevated levels of cytokines, serotonin and tryptophan in mucosal biopsies [113]. However, hyposerotonemia, lower synthesis of 5-HT in the brain, has been reported in children with ASD [114]. According to De Theije, due to low-grade gut inflammation, EC cells, mast cells and platelets are stimulated to produce serotonin, which leads to intestinal dysmotility and the consumption of tryptophan [80]. On the other hand, tryptophan is a precursor for a number of metabolites, most notably kynurenine and serotonin. Consequently, less tryptophan is available for the synthesis of serotonin in the brain, which may explain the mood and cognitive impairment associated with ASD because a reduction in tryptophan in the diet seems to worsen autistic behaviour in adults. Dysbiosis can directly affect the availability of tryptophan to the host by decreasing the number of amino acids that are absorbed from the diet [4].

However, clinical evidence of a link between hyperserotonemia and autistic behaviours (i.e., stereotypy, social impairment) is still inconsistent [115,116]. Neither the addition of tryptophan in the diet nor selective serotonin reuptake inhibitor (SSRI, i.e., that increases the brain’s levels of serotonin) administration in ASD children have been proven to be effective treatments.

## 6. Gut-Microbial Metabolites Pathway

Gut bacteria produce toxins, metabolite and co-metabolites that can cross the gut-blood and blood-brain barriers, thus influencing the brain, behaviour and gut [117]. Due to the effective measurement techniques of metabolomics, it has been possible to identify alterations in various metabolites in ASD through analyses of urinary, serum and faecal samples [22,62,118]. Children with ASD have high faecal [22] and urinary [119] levels of p-cresol and its co-metabolite p-cresyl sulfate, which are phenolic compounds that are produced by bacteria and that express p-cresol-synthesizing enzymes (i.e., *C. difficile*, *Bifidobacterium*) [42]. Early exposure to p-cresol may contribute to the severity of behavioural symptoms and cognitive impairment in ASD toddlers, as well as to gut infections and GI disorders [119,120]. High levels of SCFAs have also been reported in stools of ASD children [62,117].

According to De Angelis and colleagues, children with ASD had higher levels of propionic acid and acetic acid but lower levels of butyric acid [22]. SCFAs, such as acetic acid (AA), propionic acid (PPA) and butyric acid (BA), are the fermentation end products of non-digested carbohydrates [121] and they are thought to be involved in the pathogenesis of ASD [122]. The mechanisms through which SCFAs influence the CNS include alteration of mitochondrial functions and the epigenetic modulation of ASD-associated genes. PPA, produced by ASD-associated species (i.e., *Clostridia*, *Bacteroides* and *Desulfovibrio*) is used as a preservative in the food industry [67] and has several functions, such as modulation of neurotransmitter synthesis and release [123], anti-inflammatory and antibacterial effects, and modulation of mitochondrial and lipid metabolism [121]. PPA can also induce autistic-like behaviours in rodents [117,124,125,126,127,128].

Butyric acid (BA) is another SCFA produced by the gut microbiome that modulates gut transepithelial transport and plays a role in mitochondrial function, stimulating oxidative phosphorylation and fatty acid oxidation [129]. The administration of BT may be therapeutic in several neurologic conditions, such as dementia and depression [130,131,132], possibly due to upregulating physiological stress pathways [133]. In animal models of ASD, BA positively modulates neurotransmitter gene expression by inhibiting histone deacetylase and, contrary to the action of PPA, can rescue behavioural abnormalities in mice [134,135]. In an interesting recent study, Rose and colleagues pointed to the role of BA in the mitochondrial dysfunction observed in ASD [133]. They developed a lymphoblastoid cell line (LCL) model of ASD with a subset of LCLs presenting mitochondrial dysfunction and demonstrated that BA has a positive effect on cells from both healthy and ASD children that are under physiological stress. Further in vivo studies are needed to assess the potential therapeutic effect of BA in many diseases associated with mitochondrial dysfunction, such as ASD. Children with ASD also have a dysregulated metabolism of free amino acids (FAA), which are derived from the hydrolysis of proteins and peptides [136].

The concentration of total and individual FAA in faecal samples has been found to be higher in children with autism than in healthy children and children with PDD-NOS, and these data correlate with the prevalence of proteolytic bacteria in children with autism [22]. Glutamate, which is found in the highest levels in individuals with ASD, is an amino acid that acts as a neurotransmitter in the CNS and is thought to be involved in the etiopathogenesis of neurodevelopmental disorders [137]. Conversely, lower levels of glycine, serine, threonine, alanine, histidine, and glutamyl amino acids, as well as low levels of antioxidants, were observed in children with ASD using metabolomics in urinary specimens [136]. Children with ASD have a greater urinary excretion of tryptophan, the 5-HT precursor, and its degradation fragments, which is similar to the findings associated with other neuropsychiatric conditions [138]. In addition, increased levels of 2(4-hydroxyphenyl)propionate, taurocholenate sulfate [136] and 3-(3-hydroxyphenyl)-3-hydroxypropanoic acid were detected in urinary samples of children with ASD [22,138,139], and significantly lower levels of 3-(3-hydroxyphenyl)propionate and 5-aminovalerate were also observed [136]. In particular, 3-(3-hydroxyphenyl)-3-hydroxypropanoic acid is a phenylalanine metabolite of *Clostridium* and has been demonstrated to induce autistic-like behaviours in animal models [139].

Several factors such as diet, the gut microbiome and different gene expression may explain these findings in patients with ASD. Metabolomic characterization of ASD could help to develop new diagnostic strategies as well as therapeutic approaches based on diet and restoring the physiological gut microbiota.

## 7. Dietary Interventions: What is the Evidence?

Dietary interventions in children with ASD are very popular but could be potentially harmful. Restrictive diets further limit the variety of food intake since individuals with ASD already exhibit picky eating behaviours, so restrictive diets can result in macronutrient and micronutrient deficiencies [140,141]. On the other hand, it has been demonstrated that a Mediterranean diet impacts the gut microbiota and associated metabolome as well as cardiovascular diseases and neurobehavioural health outcomes [142,143,144,145]. Thus, investigating the physiological effects of dietary interventions on the gut-brain axis is critical to understanding the different potential therapeutic diets.

Individuals with ASD seem to have decreased digestive enzyme activity (i.e., saccharolytic enzymes) and impaired protein digestion that, together with increased gut permeability, could be responsible for the elevated levels of urinary dietary peptides, altered plasma amino acid profiles and high levels of faecal putrefactive metabolites (i.e., propionic acid) reported in children with ASD [146]. These findings have arisen from the not widely accepted “opioid excess theory”, which posits that the breakdown products of dietary protein, especially casein and gluten, act as agonists of opioid receptors that can exert effects on the CNS and neurobehaviours [147]. The main premise of this theory is that autism is the result of a metabolic disorder. Peptides with opioid activity derived from dietary sources pass through an abnormally permeable intestinal membrane and enter the CNS to exert an effect on neurotransmission, as well as producing other physiologically-based symptoms. An extended gluten-free and casein-free (GF/CF) diet in patients with ASD seems to reduce urinary peptide levels and improve behaviour [148], but supporting evidence is still weak and limited [149]. In addition, GF and CF diets only reduce levels of two toxins, gluteomorphins and caseomorphins but cannot heal the mucosa or restore gut microbiota composition [150]. Studies on the efficacy of the GF/CF diet are conflicting [53,151,152]. The different outcomes may be explained by the replacement of wheat and dairy proteins with other kinds of proteins (i.e., from maize or legumes) that are also difficult to digest by a fragile gut and the increased simple carbohydrate intake, which promotes the overgrowth of toxic bacteria in the gut of children with ASD [146]. Due to the heterogeneity of the ASD subtypes, it is reasonable to assume that at least one subgroup of these children who have GI problems with increased intestinal permeability or food allergies could respond to and benefit from a GF/CF diet, resulting in improvements in both GI and neurobehavioural symptoms [153].

Improvements in seizure control and neurobehavioural symptoms have also been reported in ASD animal models and children with mild-moderate types of ASD as a result of following a ketogenic diet (KD, i.e., a high fat diet that has demonstrated beneficial effects on mitochondrial dysfunction and epilepsy) [154,155]. Unfortunately, these children also experienced adverse effects, including constipation, hypercholesterolemia with higher inflammatory risk, menstrual irregularities, and the potentially life-threatening conditions, such as vomiting and dehydration [155]. The dangerous side effects and the increase in harmful bacteria in the gut microbiota associated with the limited number of positive results after KD administration, necessitate more research in this field. The main limitations of all dietary intervention studies are the short duration of the treatment, the lack of a division into subgroups with and without GI problems and the lack of testing for compliance [146].

Probiotic administration could represent a potential better alternative to restrictive dietary interventions because, in addition to general safety and tolerance, probiotics can heal the intestinal mucosa, protect the epithelial barrier through the production of mucin and fortifying tight junctions, increase the production of digestive enzymes and antioxidants, and modulate the immune response [141].

## 8. Therapeutic Perspectives

Currently, no definitive or effective therapies for ASD exist. The approved and recommended treatments for ASD essentially include rehabilitation, educational therapy and psycho-pharmacological approaches [140]. This has led parents to look for alternative treatments without solid scientific evidence; these treatments, such as the administration of vitamins and other supplements or the adoption of elimination diets, are often expensive and potentially dangerous (i.e., a gluten-free and casein-free diet) [156]. Vitamin A (VA) supplementation may positively influence the gut microbiome, but it remains unclear whether the VA concentration is associated with autism symptoms [157]. The use of antibiotics in ASD is controversial. Children with ASD experience significantly more ear infections and use greater amounts of antibiotics than non-ASD children [157,158]. During pregnancy, antibiotic use may be a risk factor for autism [159]. Conversely, a clinical trial of 10 toddlers with ASD showed improvements in autistic behaviours in 8 of the toddlers after a short oral course of vancomycin [64].

Because of the emerging role of gut dysbiosis in ASD, research is focusing on rebalancing the gut microbiota as a possible therapeutic approach for such diseases; this approach includes oral pre-probiotic administration and FMT.

Prebiotics are non-digestible compounds (i.e., inulin, oligosaccharides) that are metabolized by the intestinal tract and support the proliferation of beneficial gut bacteria such as Lactobacilli and *Bifidobacteria* [160]. For instance, galacto-oligosaccharides (GOS) have been shown to have a bifidogenic effect in autistic and non-autistic children [161] and may suppress the neuroendocrine stress response and ameliorate attentional vigilance in healthy volunteers [162].

However, studies have mostly focused on the potential role of probiotics in the treatment of ASD. Probiotics are living non-pathogenic microorganisms that can provide health benefits in a variety of conditions, such as obesity, IBD, colorectal cancer, and neurological diseases [5,35]. Several pre-clinical studies have shown the potential beneficial effects of probiotics in treating neurological diseases [91,163,164,165,166,167,168], whose findings have been confirmed by clinical trials performed on humans [169,170].

Faecal microbiota transplant (FMT) and microbiota transfer therapy (MTT), a modified FMT protocol, have recently attracted the interest of researchers because of their effectiveness in the treatment of recurrent *Clostridium difficile* infections [171] and their promising role in the treatment of IBD [172]. FMT consists of transferring the faecal microbiota from healthy volunteers to patients with gut dysbiosis [5], and this may alleviate GI and neurobehavioural symptoms in children with ASD by rebalancing the physiological intestinal microbiota [173].

## 9. Microbiota Transfer Therapy

Microbiota transfer therapy (MTT) appears to be an alternative and promising new approach to treating gut dysbiosis in ASD. In an open-label clinical trial, Kang and colleagues developed a modified FMT protocol (MTT) that consisted of 14 days of oral vancomycin treatment followed by 12–24 h of fasting bowel cleansing and then either oral or rectal administration of standardized human gut microbiota (SHGM) for 7–8 weeks [159]. The researchers observed improvements in ASD behavioural symptoms that continued for 8 weeks after treatment ended and a good tolerance in the patients enrolled (7–16 years of age). A randomized, double blind, placebo-controlled study is required to confirm these promising results.

## 10. Probiotic Administration in Autism Spectrum Disorder Treatment

Because probiotics can reduce gut inflammation and ameliorate GI symptoms in children with IBD [174,175,176,177], it has been postulated that probiotics may reduce the inflammatory state and reduce behavioural symptoms in children with ASD [25].

The most interesting pre-clinical study of probiotics in ASD treatment was carried out by Hsiao and colleagues on the maternal immune-activation (MIA) model of ASD, in which either autistic-like behaviours or GI disorders with abnormal gut microbiota were displayed. The oral administration of *Bacteroides fragilis* (1 × 10^9^ CFU) in a pre-clinical model of MIA offspring improved gut permeability, restored intestinal microbiota and improved autistic-like behaviours [91], suggesting microbial modulation as an effective and safe treatment for ASD. Interestingly, in early midel only *Bacteroidetes* and no other species could ameliorate anxiety, stereotyping and communicatively impaired behaviours, indicating that probiotic therapy requires some specificity [35].

There is increasing evidence that maternal obesity and diabetes are linked to autism [178,179]. A maternal high-fat diet (MHFD) induced alterations in gut microbiome composition and social withdrawal in offspring [180]. Notably, the administration of *Lactobacillus reuteri*, but not other species of probiotics, could reverse social behavioural abnormalities but not repetitive behaviours and anxiety, again suggesting the specificity of effects among probiotics strains. Furthermore, *Lactobacillus reuteri* increased the oxytocin levels involved in the mesolimbic dopamine reward system, which is thought to be dysregulated in ASD [180,181].

A recent study performed on hamsters in which autistic-like behaviours were induced by PPA and clindamycin administration studied the therapeutic effect of a three-week oral treatment with a mixture of *Bifidobacteria* and *Lactobacilli* strains (ProtexinR). Clindamycin and PPA increased brain glutamate excitotoxicity in hamsters, inducing a depletion of Mg^2+^ and GABA [182]. Previous studies on animals have shown that Mg^2+^ deficiency, resulting in an excess of Ca^2+^ and glutamate, might underlie repetitive and impaired social behaviours [127,128]. Moreover, changes in the gut microbiota with the appearance of *Clostridium* species were detected in the stool samples of hamsters treated with clindamycin and PPA. ProtexinR administration was effective in rebalancing the previous microbiome and counteracting glutamate excitotoxicity [182].

Clinical trials performed on children to study the effect of probiotics on neurological diseases have mostly investigated ASD [35]. A randomized, double-blind, placebo-controlled study recruited 62 children with ASD from 3 to 16 years of age who were given *Lactobacillus plantarum* WCFS1 (daily dose, 4.5 × 10^10^ CFU) for three weeks. Children were divided into two groups: Group I received a placebo during the first feeding period (3 weeks) and a probiotic during the second (3 weeks) and vice versa for group II. The behavioural/emotional impact of both feeding regimes (probiotic and placebo) was assessed through a standardized Development Behaviour Checklist (DBC-P). No differences between the two feeding periods were reported. However, behaviour and emotional problems scores were significantly higher (*p* < 0.05) at baseline compared to those during both the probiotic and placebo periods [183].

Oral supplementation with *Lactobacillus acidophilus* twice daily for 2 months in a cohort study of 22 children (age range 4–10 years) with ASD ameliorated their ability to concentrate and fulfil orders such as following directions. Unfortunately, no effects on behavioural or emotional impairment were reported [184]. Tomova and colleagues investigated the faecal microflora of 10 children with autism by real-time polymerase chain reaction (PCR). The abundance of *Desulfovibrio* spp. in the stool and GI disturbances in autistic patients strongly correlated with the severity of autistic behaviours. Dietary supplementation with one capsule of “Children Dophilus” (3 strains of *Lactobacillus*, 2 strains of *Bifidobacterium* and one strain of *Streptococcus*) in autistic toddlers three times a day for 4 months normalized the *Bacteroidetes/Firmicutes* ratio and *Desulfovibrio* spp. abundance, and reduced TNFα levels in their stools [59]. However, no clinical assessment after probiotic administration was performed.

A randomized clinical trial was performed on 75 infants who were followed for 13 years; *Lactobacillus rhamnosus* GG was given to 40 subjects for the first six months of life and a placebo was given to the other participants. At the age of 13 years, 6 of the 35 patients in the placebo group were diagnosed with Asperger syndrome (AS) or attention deficit/hyperactivity disorder (ADHD), but none of the participants in the probiotic group were diagnosed with these conditions (*p* = 0.008). It has been postulated that *Lactobacillus* administration early in life may reduce the risk of developing further ASD or ADHD; however, more confirming studies are required. Moreover, probiotic administration could act independently from gut microbiota alterations [185].

Two case reports showed a regression of autistic symptoms after probiotic administration [186,187], which was reversed after the term of the therapy. VSL#3 (a multi-strain mixture of 10 probiotics) given daily to a boy with ASD and severe cognitive impairment could reverse his neurobehavioural and GI symptoms [187].

A survey conducted on caregivers showed that daily administration of Delpro^®^, a mixture of five probiotic strains with the immunomodulator Del-Immune V^®^ (*Lactobacillus rhamnosus* V lysate), for 21 days in children with ASD can improve multiple domains (speech, language, communication, sociability, cognitive and behaviour), as evaluated through the Autism Treatment Evaluation Checklist (ATEC) [188]. However, this is not a validated controlled clinical trial, and we cannot accurately interpret these results compared with the others.

A recent study included 30 patients with ASD between 5 and 9 years old who were given supplemented formula containing *Lactobacillus acidophilus*, *Lactobacillus rhamnosus* and *Bifidobacteria longum* for 3 months and who were then evaluated through the ATEC. The overall ATEC scores significantly decreased after probiotic treatment (*p* = 0.0001), and a significant decrease in severity of autistic symptoms was reported in the speech/language/communication categories of the ATEC (*p* < 0.0001) [189].

Interestingly, a still ongoing randomized controlled trial will determine the effect of a 6-month supplementation with Vivomixx^®^, a probiotic mixture containing one strain of *Streptococcus thermophilus DSM 24731*, three strains of *Bifidobacterium*, and four strains of *Lactobacillus*, on children with ASD with or without GI symptoms. The posology protocol includes 2 doses per day for the first month and one dose per day in the following 5 months of treatment. A clinical assessment will be performed on the groups treated with probiotics (with and without GI symptoms) and with placebo (with and without GI symptoms)to evaluate the possible improvements in GI symptoms, cognitive and language impairment, autism severity and behavioural symptoms. Moreover, changes in plasmatic, urinary and faecal biomarkers related to dysbiosis and neurophysiological patterns will be analysed after probiotic treatment [140]. Conversely, other investigations have shown that probiotics could not reverse dysbiosis and that other interventions may be studied (i.e., engineered bacteria) [190].

In summary, the possible positive effect of probiotic treatment on neurobehavioural symptoms in children with ASD is still controversial. The main limits of the clinical trials mentioned above are the lack of homogeneity in the patients enrolled (i.e., age, presence/absence of GI symptoms, different therapies and dietary habits), small sample sizes, distinct duration and number of probiotics administered as well as different scales used for autistic symptom evaluation after treatment.

Table 1 summarizes the main clinical trials performed on the effects of probiotics on children with ASD.

## 11. Conclusions

Alterations in gut microbiome composition have been confirmed in children with ASD. However, the available data do not allow us to define a characteristic and unique profile of ASD, and some results are conflicting, perhaps due to the heterogeneity of the patients enrolled. Bowel dysfunctions are frequent in ASD children and may correlate with the severity of autism but are not present in all cases of ASD. This finding suggests that we should consider two subtypes of ASD, with different grade of inflammation playing a possible role in the etiopathogenesis of ASD associated with GI comorbidities. This dichotomy of the presence or not of gastrointestinal disorders in ASD patients may explain the non-consensual results of studies on dysbiosis and ASD [191,192]. Among all the therapeutic approaches to autism, the potential usefulness of probiotics in reducing autistic symptoms has been investigated. Clinical trials with children have suggested promising results, but they are still limited and lack safety and tolerability evaluations. In addition, several limitations exist in the methods of microbiota analysis, reducing the possibility of targeted metabolomics approach [193]. Further well-designed, randomized, placebo-controlled clinical trials are required to validate the effectiveness of probiotics in ASD treatment and to identify the appropriate strains, dose and timing of treatment.

## Figures and Tables

**Table 1 nutrients-11-00521-t001:** Main clinical trials performed on the effects of probiotics on children with autism spectrum disorder (ASD).

Authors	Study design	Treatment	Effects
Parracho et al. [183]	Randomized, double-blind, placebo controlled study on children with ASD from 3–16 years old	*L. plantarum* WCFS1 vs. placebo for 3 weeks	Improvement of disruptive antisocial behaviours, anxiety and communication problems in probiotic arm
Kaluzna-Czaplinska et al. [184]	Cohort study of children with ASD from 4–10 years old	*L. acidophilus* strain Rosell-11 for 2 months	Improvement in their ability to concentrate and fulfil orders, with no impact on behavioural responses to other people’s emotions or eye contact
Partty et al. [185]	Randomized trial, placebo controlled study on infants followed for 13 years	*L. rhamnosus* GG vs. placebo for the first 6 months of life	At the age of 13 years, 6 out of 35 (17.1%) children in the placebo group were diagnosed with ASD or attention-deficit/hyperactivity disorder, but none in the probiotic group were
Tomova et al. [59]	Real-time PCR on faecal samples of 10 children with autism before and after probiotic administration	A mixture of *Lactobacilli*, *Bifidobacteria* and *Streptococci* given 3 times a day for 4 months	Normalization of Bacteroides/Firmicutes ratio and Desulfovibrio spp. abundance. No clinical assessments were performed.
Blades M. [186]	Case report of a 6-year-old child with ASD	Probiotic administration for two months	Improvements in school records and attitude towards food reversed after probiotics discontinuation
Grossi et al. [187]	Case report	A mixture of *Bifidobacteria*, *Lactobacilli* and *Streptococci* given daily for 4 weeks	Reduction of neurobehavioural and gastrointestinal symptoms
West et al. [188]	Cohort study of 33 children with ASD	Delpro^®^ (*Lactocillus acidophilus*, *Lactobacillus casei*, *Lactobacillus delbruecki*, *Bifdobacteria longum*, *Bifidobacteria bifdum*) and Del-Immune V^®^ (*Lactobacillus rhamnosus* V lysate), for 21 days	Decrease in ATEC (Autism Treatment evaluation checklist) score in 88% of children
Shaaban et al. [189]	Cohort study of 30 children with ASD from 5 to 9 years old	*Lactobacillus acidophilus*, *Lactobacillus rhamnosus* and *Bifidobacteria longum* for 3 months	Improvements in gastrointestinal problems and decrease in ATEC score
Santocchi et al. [140]	Randomized, placebo controlled trial of a group of 100 children with ASD. They were classified as belonging to the GI group or the non-GI group, blind randomized 1:1 to regular diet with probiotic or with placebo for 6 months.	*Vivomixx*^®^, a probiotic mixture containing one strain of *Streptococcus thermophilus*, three strains of *Bifidobacterium*, and four strains of *Lactobacillus*	The study is still ongoing. All participants will be assessed at baseline, after three months and after 6 months in order to evaluate gastrointestinal and neurobehavioural changes.

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
