# Peer review of "Autism Spectrum Disorders and the Gut Microbiota"

_nutrients, 2019, doi:10.3390/nu11030521_

Reviewer 1 Report

Dear Authors,

Having read your manuscript, I found it to be engaging and to provide an interesting angle on the relationship between the gastrointestinal microbiota and autism. Please find my comments below: 

Introduction section - From line 30 to line 111

Your definition of microbiota doesn't convey clearly enough how its role in health and disease, which is as crucial as is complex. Ample evidence from both human and animal studies describes the impact of dietary interventions on the idiosyncrasies the gut microbiota. Tens of trillions of microorganisms inhabit our gastrointestinal system in quantities and diversity increasing from stomach to small intestine to colon. 

I believe it would be appropriate to include a brief paragraph to explain how - under natural conditions -  intestinal bacteria share their habitat with a dynamic community of viruses, protozoa, helminths and fungi many of which exhibit parasitic behaviour. Every individual possesses a unique community of microorganisms that interacts with their human host through immune, neuroendocrine and neural pathways, thereby casting local as well as systemic effects on the host’s health as well as their disease risks. Moreover, alterations in normal commensal gut microbiota result in an increase in pathogenic microbes that deranges both microbial and host homeostasis. This microbial imbalance is known as dysbiosis and it has been widely reported as a key contributor to the multiple system dysregulation observed in the pathogenesis of cardiovascular metabolic, neuroimmune and neurobehavioural conditions. 

I believe the papers below would be of interest to you and can help you include a more comprehensive angle, as per my notes above. I trust you will find them interesting:  

1.         Backhed, F.; Ley, R.E.; Sonnenburg, J.L.; Peterson, D.A.; Gordon, J.I. Host-bacterial mutualism in the human intestine. Science2005, 307, 1915-1920.10.1126/science.1104816

2.         Backhed, F.; Roswall, J.; Peng, Y.; Feng, Q.; Jia, H.; Kovatcheva-Datchary, P. Dynamics and stabilization of the human gut microbiome during the first year of life. Cell Host Microbe 2015, 17.10.1016/j.chom.2015.05.012

3.         Sekirov, I.; Russell, S.L.; Antunes, L.C.; Finlay, B.B. Gut microbiota in health and disease. Physiol Rev 2010, 90, 859-904.10.1152/physrev.00045.2009

4.         Brown, E.M.; Sadarangani, M.; Finlay, B.B. The role of the immune system in governing host-microbe interactions in the intestine. Nat Immunol 2013, 14, 660-667.10.1038/ni.2611

5.         Marzano, V.; Mancinelli, L.; Bracaglia, G.; Del Chierico, F.; Vernocchi, P.; Di Girolamo, F.; Garrone, S.; Tchidjou Kuekou, H.; D'Argenio, P.; Dallapiccola, B., et al."Omic" investigations of protozoa and worms for a deeper understanding of the human gut "parasitome". PLoS Negl Trop Dis 2017, 11, e0005916.10.1371/journal.pntd.0005916

7.         Jenkins, T.P.; Rathnayaka, Y.; Perera, P.K.; Peachey, L.E.; Nolan, M.J.; Krause, L.; Rajakaruna, R.S.; Cantacessi, C. Infections by human gastrointestinal helminths are associated with changes in faecal microbiota diversity and composition. PLoS One 2017, 12, e0184719.10.1371/journal.pone.0184719

8.         Leung, J.M.; Graham, A.L.; Knowles, S.C.L. Parasite-microbiota interactions with the vertebrate gut: Synthesis through an ecological lens. Front Microbiol 2018, 9, 843.10.3389/fmicb.2018.00843

9.         Chabe, M.; Lokmer, A.; Segurel, L. Gut protozoa: Friends or foes of the human gut microbiota? Trends Parasitol 2017, 33, 925-934.10.1016/jNaN.2017.08.005

Dietary interventions: what is the evidence? From line 428 to 466

Given the importance of the link between diet and gut microbiota, this section seems to lack a fair amount of detail. I would have expected to find some notes on the evidence of the effect of a Mediterranean diet pattern and wider microbial diversity, which is also seen associated with a variety of neurobehavioural health outcomes. 

I have included some papers that you might find provide good sources of data to include in this section. 

1.    Estruch, R.; Ros, E.; Salas-Salvado, J.; Covas, M.I.; Corella, D.; Aros, F.; Gomez-Gracia, E.; Ruiz-Gutierrez, V.; Fiol, M.; Lapetra, J., et al.Primary prevention of cardiovascular disease with a mediterranean diet. N Engl J Med 2013, 368, 1279-1290.10.1056/NEJMoa1200303

2.     Toribio-Mateas, M. Harnessing the power of microbiome assessment tools as part of neuroprotective nutrition and lifestyle medicine interventions. Microorganisms 2018, 6, 35

3.      Guasch-Ferre, M.; Hu, F.B.; Ruiz-Canela, M.; Bullo, M.; Toledo, E.; Wang, D.D.; Corella, D.; Gomez-Gracia, E.; Fiol, M.; Estruch, R., et al.Plasma metabolites from choline pathway and risk of cardiovascular disease in the predimed (prevention with mediterranean diet) study. J Am Heart Assoc 2017, 6.10.1161/jaha.117.006524

4.     De Filippis, F.; Pellegrini, N.; Vannini, L.; Jeffery, I.B.; La Storia, A.; Laghi, L.; Serrazanetti, D.I.; Di Cagno, R.; Ferrocino, I.; Lazzi, C., et al.High-level adherence to a mediterranean diet beneficially impacts the gut microbiota and associated metabolome. Gut 2016, 65, 1812-1821.10.1136/gutjnl-2015-309957

All best wishes.

 Author Response

Dear Authors,

Having read your manuscript, I found it to be engaging and to provide an interesting angle on the relationship between the gastrointestinal microbiota and autism. Please find my comments below: 

Re: Thank you very much for your suggestions. We revised the manuscript accordingly.

Introduction section - From line 30 to line 111

Your definition of microbiota doesn't convey clearly enough how its role in health and disease, which is as crucial as is complex. Ample evidence from both human and animal studies describes the impact of dietary interventions on the idiosyncrasies the gut microbiota. Tens of trillions of microorganisms inhabit our gastrointestinal system in quantities and diversity increasing from stomach to small intestine to colon. 

Re: We clarified the microbiota role in health and disease (p. 2).

I believe it would be appropriate to include a brief paragraph to explain how - under natural conditions -  intestinal bacteria share their habitat with a dynamic community of viruses, protozoa, helminths and fungi many of which exhibit parasitic behaviour. Every individual possesses a unique community of microorganisms that interacts with their human host through immune, neuroendocrine and neural pathways, thereby casting local as well as systemic effects on the host’s health as well as their disease risks. Moreover, alterations in normal commensal gut microbiota result in an increase in pathogenic microbes that deranges both microbial and host homeostasis. This microbial imbalance is known as dysbiosis and it has been widely reported as a key contributor to the multiple system dysregulation observed in the pathogenesis of cardiovascular metabolic, neuroimmune and neurobehavioural conditions.

Re: The suggested paragraph has been added (p. 2).

I believe the papers below would be of interest to you and can help you include a more comprehensive angle, as per my notes above. I trust you will find them interesting:  

Re: The suggested references have been included (pp. 2, 14, 15).

1.         Backhed, F.; Ley, R.E.; Sonnenburg, J.L.; Peterson, D.A.; Gordon, J.I. Host-bacterial mutualism in the human intestine. Science2005, 307, 1915-1920.10.1126/science.1104816

2.         Backhed, F.; Roswall, J.; Peng, Y.; Feng, Q.; Jia, H.; Kovatcheva-Datchary, P. Dynamics and stabilization of the human gut microbiome during the first year of life. Cell Host Microbe 2015, 17.10.1016/j.chom.2015.05.012

3.         Sekirov, I.; Russell, S.L.; Antunes, L.C.; Finlay, B.B. Gut microbiota in health and disease. Physiol Rev 2010, 90, 859-904.10.1152/physrev.00045.2009

4.         Brown, E.M.; Sadarangani, M.; Finlay, B.B. The role of the immune system in governing host-microbe interactions in the intestine. Nat Immunol 2013, 14, 660-667.10.1038/ni.2611

5.         Marzano, V.; Mancinelli, L.; Bracaglia, G.; Del Chierico, F.; Vernocchi, P.; Di Girolamo, F.; Garrone, S.; Tchidjou Kuekou, H.; D'Argenio, P.; Dallapiccola, B., et al."Omic" investigations of protozoa and worms for a deeper understanding of the human gut "parasitome". PLoS Negl Trop Dis 2017, 11, e0005916.10.1371/journal.pntd.0005916

7.         Jenkins, T.P.; Rathnayaka, Y.; Perera, P.K.; Peachey, L.E.; Nolan, M.J.; Krause, L.; Rajakaruna, R.S.; Cantacessi, C. Infections by human gastrointestinal helminths are associated with changes in faecal microbiota diversity and composition. PLoS One 2017, 12, e0184719.10.1371/journal.pone.0184719

8.         Leung, J.M.; Graham, A.L.; Knowles, S.C.L. Parasite-microbiota interactions with the vertebrate gut: Synthesis through an ecological lens. Front Microbiol 2018, 9, 843.10.3389/fmicb.2018.00843

9.         Chabe, M.; Lokmer, A.; Segurel, L. Gut protozoa: Friends or foes of the human gut microbiota? Trends Parasitol 2017, 33, 925-934.10.1016/jNaN.2017.08.005

 Dietary interventions: what is the evidence? From line 428 to 466

Given the importance of the link between diet and gut microbiota, this section seems to lack a fair amount of detail. I would have expected to find some notes on the evidence of the effect of a Mediterranean diet pattern and wider microbial diversity, which is also seen associated with a variety of neurobehavioural health outcomes. 

Re: Some notes have been included as suggested (p. 10).

I have included some papers that you might find provide good sources of data to include in this section. 

Re: Suggested references have been included (pp. 10 and 22).

1.    Estruch, R.; Ros, E.; Salas-Salvado, J.; Covas, M.I.; Corella, D.; Aros, F.; Gomez-Gracia, E.; Ruiz-Gutierrez, V.; Fiol, M.; Lapetra, J., et al.Primary prevention of cardiovascular disease with a mediterranean diet. N Engl J Med 2013, 368, 1279-1290.10.1056/NEJMoa1200303

2.     Toribio-Mateas, M. Harnessing the power of microbiome assessment tools as part of neuroprotective nutrition and lifestyle medicine interventions. Microorganisms 2018, 6, 35

3.      Guasch-Ferre, M.; Hu, F.B.; Ruiz-Canela, M.; Bullo, M.; Toledo, E.; Wang, D.D.; Corella, D.; Gomez-Gracia, E.; Fiol, M.; Estruch, R., et al.Plasma metabolites from choline pathway and risk of cardiovascular disease in the predimed (prevention with mediterranean diet) study. J Am Heart Assoc 2017, 6.10.1161/jaha.117.006524

4.     De Filippis, F.; Pellegrini, N.; Vannini, L.; Jeffery, I.B.; La Storia, A.; Laghi, L.; Serrazanetti, D.I.; Di Cagno, R.; Ferrocino, I.; Lazzi, C., et al.High-level adherence to a mediterranean diet beneficially impacts the gut microbiota and associated metabolome. Gut 2016, 65, 1812-1821.10.1136/gutjnl-2015-309957

Reviewer 2 Report

Fattorusso et al.

Autism spectrum disorders and the gut microbiota

In this manuscript, Fattorusso et al. reviewed the recent literature dealing with autism spectrum disorders (ASD) and gut microbiota, both in pre-clinical and clinical studies. This is a well-written and exhaustive review of the literature, which is very abundant in this particular field. It would be therefore useful that the authors precise in their introduction the originality of their approach and their contribution to the understanding of the complex relationship between ASD and gut microbiota (i.e. the necessity to consider the presence or not of gastrointestinal disorders….), as some very recent reviews dealing with exactly the same topic have just issued early 2019 (Liu et al, Altered composition and function of intestinal microbiota in autism spectrum disorders: a systematic review, Translational Psychiatry, 2019, 9:43; Fetissov et al, Neuropeptides in the microbiota-brain axis and feeding behaviour in autism spectrum disorder, Nutrition, 2019, 61:43-48). The authors must also indicate at the end of the introduction section (L.69-70) when the PubMed research was performed.

The word “nexus” appears several times in the manuscript (L.15 in the abstract, L.67 in the introduction, L.305 and L.337) but its signification is not explained and as it signifies “gap junction” in its biological sense, it would be more suitable to use another simpler and less ambiguous word such as “link”.

The review is well organized in a logical and didactic progression, presenting the general context and the methodology used in the introduction section, then the gut microbiota, the state of the art on dysbiosis in ASD, the gastrointestinal disorders (GI) disorders in ASD and the “inflammation hypothesis”. At this stage, a figure recapitulating the two latter paragraphs (dysbiosis and ASP and GI disorders and the inflammation hypothesis) is missing and will help the reader to have a clearer picture of the relationships between dysbiosis, GI disorders, inflammation and ASD, and the possible links between them. It is a strength of this review to introduce the dichotomy of the presence or not the GI disorders in ASD patients that may explain the non-consensual results on dysbiosis and ASD, and it would be interesting to discuss what is known on the subject.

The title of following paragraph (“The microbiota, autism spectrum disorders and the gut-brain axis”) is not accurate as it presents the gut-brain axis with no reference (but two, L. 273 and L.299) on ASD. This paragraph may not be mandatory since many of the pieces of information brought here have already been mentioned. Furthermore, the authors chose to present “five of the main known pathways of communication between the gut and the brain” but some are redundant, and they are not the one usually pointed out (which are usually nutrients, metabolites, hormones, neurotransmitters directly or via the vagus nerve rather than gut epithelial permeability, immunological pathways, ENS, then metabolites, bacterial neurotransmitters…) and they deal indifferently with functions, molecules… Recalling a recent review on microbiota-gut-brain axis would be enough.

Then, the reason why the authors focus on the role of serotonin, with a whole paragraph dealing with it, is not clear at all and should be better explained (ref 96). In the corresponding paragraph, there are discrepancies which should be explained i.e.: germ free animals have higher levels of monoamines (including serotonin) and less anxiety-like behaviours than conventional mice but ASD children also display hyperserotonemia, which seems contradictory… This should be discussed. Furthermore, the sentence L.303-304 saying that “the production of certain neurotransmitters is crucial for the subsistence of gut microorganisms in stressful situations” is not supported by the cited literature and has to be explained or corrected. Globally, this paragraph on serotonin is too elliptic and the rational for focusing on serotonin in ASD and the suspected mechanisms have to be clarified.

The paragraph “Gut-microbial metabolite pathway” would be more logical following the one on “Dysbiosis in autism spectrum disorder” earlier in the document, as it contributes to explain the link between dysbiosis and ASD.

The following paragraphs should also be re-organized as as they stands, the probiotics topic is redundant: I would suggest to organize them as follow:

Dietary interventions: what is the evidence? / Therapeutic perspectives / Probiotic administration in autism spectrum disorder treatment. I would insert the last paragraph (“Microbiota transfer therapy”) at the end of the paragraph “Therapeutic perspectives” as it is short and partly redundant with what is said at the end of the “Therapeutic perspectives” paragraph.

For the detailed review, see below.

The gut microbiota

L. 100-103: It does not seem relevant to introduce specific pieces of information on the consequences of maternal nutrition or post-natal nutrition (breast-feeding) on the risk of ASD in this general paragraph dealing with gut microbiota and the way it colonizes the gut. The pieces of information dealing with ASD will be more appropriate in the following paragraph on dysbiosis and ASD.

Dysbiosis in autism spectrum disorder

L. 169-181: Please better explain why there is a need to investigate further duodenum and oral microbiomes. The duodenum microbiota is quite anecdotic. Actually, the mentioned studies did not find any differences in oral and duodenal microbiota between ASD patients and healthy controls. This paragraph could be removed.

Gastrointestinal abnormalities in autism spectrum disorders and the “inflammation hypothesis”

There is a contradiction between “most children with ASD have GI disorders” (L.199) and “40% of ASD patients complain of GI symptoms” (L201-202): please mitigate the first sentence.

L. 227: Please explain briefly what the AGE-RAGE axis is.

L. 237: “but dysbiosis may also be the result of primary barrier damage”: please explain.

L. 250: explain briefly how zonulin modulates gut permeability (increasing or decreasing it?).

The microbiota, autism spectrum disorder and the gut-brain axis

L. 313-320: This paragraph is not clear at all. Explain the relationship between serotonin transporter variants (SERT), increased number of EC cells and the 5-HT4 receptor and its agonist prucalopride.

L. 324: “impaired intestinal production of 5-HT”: please precise if it is a decrease or an increase.

L. 335-336: This sentence is too elliptic; please explain how dysbiosis can decrease the amount of amino acids that are absorbed from the diet, affecting the availability of tryptophan to the host.

L. 338-340: Please explain briefly why SSRI could be a treatment for ASD children.

Dietary interventions: what is the evidence?

L.429: “expensive”? in which way, please explain.

L. 438-440: please explain briefly the “opioid excess theory” and why an excess of agonists of opioid receptors can exert a negative effect on CNS and neurobehaviours.

Probiotic administration in autism spectrum disorder treatment

L. 468-472: can be removed, already said earlier.

L. 483-487: precise in Humans or the pre-clinical model used.

Minors:

L.122-123: in ASD children? Please specify.

L. 176: precise “oral” bacterial diversity

L. 302: “conventional” instead of “non GF”

L. 306: “5-HT” instead of “5-HTP”

L. 313: defect? Do you mean hypersecretion?

L. 341: “metabolite” instead of “metabolites”

L. 357: use the abbreviation defined earlier “PPA” instead of “Propionic acid”

L. 362: Butyric acid abbreviation has been defined earlier as BA: please be consistent and remplace “BT” by “BA” on L.362, 364, 366, 369, 371, 373…

L.422: FMT: this abbreviation has already been defined earlier (L.410)

L. 489: PPA: use directly the abbreviation which has been defined earlier

L. 515: “and” instead of “as well”

Author Response

Autism spectrum disorders and the gut microbiota

In this manuscript, Fattorusso et al. reviewed the recent literature dealing with autism spectrum disorders (ASD) and gut microbiota, both in pre-clinical and clinical studies. This is a well-written and exhaustive review of the literature, which is very abundant in this particular field. It would be therefore useful that the authors precise in their introduction the originality of their approach and their contribution to the understanding of the complex relationship between ASD and gut microbiota (i.e. the necessity to consider the presence or not of gastrointestinal disorders….), as some very recent reviews dealing with exactly the same topic have just issued early 2019 (Liu et al, Altered composition and function of intestinal microbiota in autism spectrum disorders: a systematic review, Translational Psychiatry, 2019, 9:43; Fetissov et al, Neuropeptides in the microbiota-brain axis and feeding behaviour in autism spectrum disorder, Nutrition, 2019, 61:43-48). The authors must also indicate at the end of the introduction section (L.69-70) when the PubMed research was performed.

Re: Thank you very much for your comments. We revised the manuscript according to your recommendation (p. 2). Moreover, the two suggested references have been included (pp. 17 and 23).

The word “nexus” appears several times in the manuscript (L.15 in the abstract, L.67 in the introduction, L.305 and L.337) but its signification is not explained and as it signifies “gap junction” in its biological sense, it would be more suitable to use another simpler and less ambiguous word such as “link”.

Re: The word “link” replaces “nexus” (pp. 1 and 8).

The review is well organized in a logical and didactic progression, presenting the general context and the methodology used in the introduction section, then the gut microbiota, the state of the art on dysbiosis in ASD, the gastrointestinal disorders (GI) disorders in ASD and the “inflammation hypothesis”. At this stage, a figure recapitulating the two latter paragraphs (dysbiosis and ASP and GI disorders and the inflammation hypothesis) is missing and will help the reader to have a clearer picture of the relationships between dysbiosis, GI disorders, inflammation and ASD, and the possible links between them. It is a strength of this review to introduce the dichotomy of the presence or not the GI disorders in ASD patients that may explain the non-consensual results on dysbiosis and ASD, and it would be interesting to discuss what is known on the subject.

Re: We gave strength at the dichotomy that you suggested (p. 17).

The title of following paragraph (“The microbiota, autism spectrum disorders and the gut-brain axis”) is not accurate as it presents the gut-brain axis with no reference (but two, L. 273 and L.299) on ASD. This paragraph may not be mandatory since many of the pieces of information brought here have already been mentioned. Furthermore, the authors chose to present “five of the main known pathways of communication between the gut and the brain” but some are redundant, and they are not the one usually pointed out (which are usually nutrients, metabolites, hormones, neurotransmitters directly or via the vagus nerve rather than gut epithelial permeability, immunological pathways, ENS, then metabolites, bacterial neurotransmitters…) and they deal indifferently with functions, molecules… Recalling a recent review on microbiota-gut-brain axis would be enough.

Re: The paragraph has been included in the previous text and its lenght has been reduced as recommended (pp. 6 and 7).

Then, the reason why the authors focus on the role of serotonin, with a whole paragraph dealing with it, is not clear at all and should be better explained (ref 96). In the corresponding paragraph, there are discrepancies which should be explained i.e.: germ free animals have higher levels of monoamines (including serotonin) and less anxiety-like behaviours than conventional mice but ASD children also display hyperserotonemia, which seems contradictory… This should be discussed. Furthermore, the sentence L.303-304 saying that “the production of certain neurotransmitters is crucial for the subsistence of gut microorganisms in stressful situations” is not supported by the cited literature and has to be explained or corrected. Globally, this paragraph on serotonin is too elliptic and the rational for focusing on serotonin in ASD and the suspected mechanisms have to be clarified.

Re: This section has been shortened and clarified (pp. 7 and 8).

The paragraph “Gut-microbial metabolite pathway” would be more logical following the one on “Dysbiosis in autism spectrum disorder” earlier in the document, as it contributes to explain the link between dysbiosis and ASD.

Re: Due to the revision of the other paragraphs, we did not think that further variations in the order of this section are needed.

The following paragraphs should also be re-organized as as they stands, the probiotics topic is redundant: I would suggest to organize them as follow:

Dietary interventions: what is the evidence? / Therapeutic perspectives / Probiotic administration in autism spectrum disorder treatment. I would insert the last paragraph (“Microbiota transfer therapy”) at the end of the paragraph “Therapeutic perspectives” as it is short and partly redundant with what is said at the end of the “Therapeutic perspectives” paragraph.

Re: Paragraphs have been re-organized as recommended (pp. 9-15).

For the detailed review, see below.

The gut microbiota

L. 100-103: It does not seem relevant to introduce specific pieces of information on the consequences of maternal nutrition or post-natal nutrition (breast-feeding) on the risk of ASD in this general paragraph dealing with gut microbiota and the way it colonizes the gut. The pieces of information dealing with ASD will be more appropriate in the following paragraph on dysbiosis and ASD.

Re: The text has been revised as suggested (p. 3).

Dysbiosis in autism spectrum disorder

L. 169-181: Please better explain why there is a need to investigate further duodenum and oral microbiomes. The duodenum microbiota is quite anecdotic. Actually, the mentioned studies did not find any differences in oral and duodenal microbiota between ASD patients and healthy controls. This paragraph could be removed.

Re: The sentence has been removed as suggested (p. 5).

Gastrointestinal abnormalities in autism spectrum disorders and the “inflammation hypothesis”

There is a contradiction between “most children with ASD have GI disorders” (L.199) and “40% of ASD patients complain of GI symptoms” (L201-202): please mitigate the first sentence.

Re: Revised as suggested (p. 5).

L. 227: Please explain briefly what the AGE-RAGE axis is.

Re: Clarified (p. 6).

L. 237: “but dysbiosis may also be the result of primary barrier damage”: please explain.

Re: Clarified (p. 6).

L. 250: explain briefly how zonulin modulates gut permeability (increasing or decreasing it?).

Re: Clarified (p. 6).

The microbiota, autism spectrum disorder and the gut-brain axis

L. 313-320: This paragraph is not clear at all. Explain the relationship between serotonin transporter variants (SERT), increased number of EC cells and the 5-HT4 receptor and its agonist prucalopride.

Re: Clarified (pp. 7 and 8).

L. 324: “impaired intestinal production of 5-HT”: please precise if it is a decrease or an increase.

Re: Clarified (p. 8).

L. 335-336: This sentence is too elliptic; please explain how dysbiosis can decrease the amount of amino acids that are absorbed from the diet, affecting the availability of tryptophan to the host.

Re: Clarified (p. 8).

L. 338-340: Please explain briefly why SSRI could be a treatment for ASD children.

Re: Done (p. 8).

Dietary interventions: what is the evidence?

L.429: “expensive”? in which way, please explain.

Re: Clarified (p. 10).

L. 438-440: please explain briefly the “opioid excess theory” and why an excess of agonists of opioid receptors can exert a negative effect on CNS and neurobehaviours.

Re: Clarified (p. 10).

Probiotic administration in autism spectrum disorder treatment

L. 468-472: can be removed, already said earlier.

Re: Removed (p. 10).

L. 483-487: precise in Humans or the pre-clinical model used.

Re: Clarified (p. 11).

Minors:

L.122-123: in ASD children? Please specify.

Re: Done (p. 3).

L. 176: precise “oral” bacterial diversity

Re: Clarified (p. 7).

L. 302: “conventional” instead of “non GF”

Re: Clarified (p. 7).

L. 306: “5-HT” instead of “5-HTP”

Re: Corrected (p. 7).

L. 313: defect? Do you mean hypersecretion?

Re: Clarified (p. 7).

L. 341: “metabolite” instead of “metabolites”

Re: Corrected (p. 8).

L. 357: use the abbreviation defined earlier “PPA” instead of “Propionic acid”

Re: Corrected (p. 8).

L. 362: Butyric acid abbreviation has been defined earlier as BA: please be consistent and remplace “BT” by “BA” on L.362, 364, 366, 369, 371, 373…

Re: Done (p. 8).

L.422: FMT: this abbreviation has already been defined earlier (L.410)

Re: Revised (p. 9).

L. 489: PPA: use directly the abbreviation which has been defined earlier

Re: Done (p. 11).

L. 515: “and” instead of “as well”

Re: Done (p. 11).

Reviewer 3 Report

There have been a proliferation of these kinds of reviews in recent years, while I have objection with the content and liked the vancomycin pies, it is not terribly novel. For instance # HPHPA has been recently studied in Bioanalysis 2018 Jaskiw  Obrenovich et al., but no reference was made and this was finding that demonstrated the compound in human CSF. I would add more current references and some greater detail abut the mechanisms of autism and ASD. I would suggest they search for markers of ASD and autism and add those they find. This would help the readers. A  plagiarism check program may have detected plagiarized pieces of this submission. I suggest the authors do a full scan and report back.

Author Response

There have been a proliferation of these kinds of reviews in recent years, while I have objection with the content and liked the vancomycin pies, it is not terribly novel. For instance # HPHPA has been recently studied in Bioanalysis 2018 Jaskiw  Obrenovich et al., but no reference was made and this was finding that demonstrated the compound in human CSF. I would add more current references and some greater detail abut the mechanisms of autism and ASD. I would suggest they search for markers of ASD and autism and add those they find. This would help the readers. A  plagiarism check program may have detected plagiarized pieces of this submission. I suggest the authors do a full scan and report back.

Re: Thank you very much for your comments. We have included several new references in our manuscript (pp. 2, 14, 15, 20, and 23).